# Enhancing tuberculosis treatment support: A thematic analysis of interactive messages in a digital adherence technology trial to identify needs, challenges, and strategies for improvement

Javier Roberti[1,2], Priscilla Carmiol-Rodríguez[3], Enrique Chan-Liang[3], Omar Alfonso Aguilar-Vidrio[3], Agnes A. Suyanto[3], Jennifer Sprecher[3], Fernando Rubinstein[1], Sarah Iribarren[3]*

1 Institute for Clinical Effectiveness and Health Policy (IECS), Buenos Aires, Argentina, 2 National Scientific and Technical Research Council (CIESP/CONICET), Buenos Aires, Argentina, 3 Biobehavioral Nursing and Health Informatics, University of Washington, Seattle, Washington, United States of America

* sjiribar@uw.edu

## Abstract

### Background

Digital adherence technologies (DATs) offer promising solutions for monitoring and supporting adherence to complex treatment regimens, including tuberculosis (TB) treatment. However, better understanding is needed of how users engage with DATs and how engagement influences their effectiveness, particularly in real-world settings and among underserved populations.

### Objective

Assess user engagement with the Companion app within the Tuberculosis Treatment Support Tools (TB-TST) intervention and explore the types of interactions that promote treatment adherence and address patients' needs.

### Methods

Secure message threads (N = 255) were analyzed as part of a four-site pragmatic clinical trial evaluating the effectiveness of the TB-TST intervention. Pragmatic thematic analysis was applied to chat logs between patients and treatment supporters (TSs) to identify key themes related to application utilization, treatment adherence, technical challenges, and communication patterns.

**Data availability statement:** The data underlying this study contain potentially identifying and sensitive patient information. As the dataset comprises interactive SMS communications between patients and treatment supporters, it includes personal and health-related content such as names, phone numbers, medical history, adherence status, and narrative descriptions of personal circumstances. These exchanges often involve sensitive disclosures about health conditions, social and emotional wellbeing, and treatment experiences. Because of this, public sharing of the de-identified dataset is restricted in order to protect participant confidentiality and comply with ethical and legal obligations. These restrictions have been imposed by the Institutional Review Boards (IRBs) that approved the study. All participants provided written informed consent but not for public sharing of their communication data. Researchers who are interested in accessing the minimal dataset for replication or secondary analysis may submit a formal request to the Human Subjects Division, University of Washington via email (hsdinfo@uw.edu), telephone (+1 (206) 543-0098), postal mail (Box 359470, Seattle, WA, USA), or from their website (https://www.washington.edu/research/hsd/). Requests will be evaluated in accordance with ethical guidelines and data protection policies to ensure participant privacy and appropriate use of the data. Interested researchers should include a brief description of their intended use, a data security plan, and documentation of IRB approval or exemption from their home institution when requesting access to the dataset.

**Funding:** This study was funded by the United States National Institute of Health, National Institute of Allergy and Infectious Diseases (R01AI147129: Iribarren, Rubinstein), and is building on the development research funded by the National Institute of Nursing Research (K23NR017210: Iribarren).

**Competing interests:** The authors have declared that no competing interests exist.

## Results

Key themes that emerged included: experiences of living with TB; COVID-19 pandemic impact; symptom guidance, healthcare coordination, and treatment and intervention adherence and technical issues. Patients frequently sought advice on how to integrate TB treatment into their daily lives, manage side effects, and cope with emotional distress. The communication styles of TSs varied, with personalized and empathetic interactions leading to better patient acceptance. Technical issues and difficulties accessing phone credit and connectivity challenges were barriers to intervention adherence. Recommendations from this analysis emphasize the need for personalized communication, streamlined adherence reporting, timely symptom guidance, expanded mental health support, and improved healthcare coordination to strengthen the intervention.

## Conclusion

This study highlights the importance of personalized, empathetic communication, and reliable technology to enhance the effectiveness of DATs. Future DAT solutions should prioritize technological reliability and the human elements of care. Interventions must be adaptable, user-friendly, and capable of addressing diverse patient needs, including offering emotional support and empowering individuals with limited digital literacy.

## Introduction

High-quality, patient-centered healthcare relies on effective communication between patients and healthcare providers. Secure messaging (SM) has emerged as a vital tool for facilitating such communication between visits and is increasingly recognized as a cornerstone of patient-centered care [1]. For individuals managing chronic diseases or infectious diseases requiring long and complex treatment such as tuberculosis (TB), communication barriers can impede treatment adherence and negatively affect health outcomes. Digital adherence technologies (DATs) offer a potential solution by facilitating real-time communication and support between patients and treatment supporters [2–5].

DATs encompass a range of tools designed to monitor and promote medication adherence, including medication event monitors (MEMS), video-observed therapy (VOT), ingestible sensors, and two-way SMS reminders [6,7]. While these technologies have shown promise in improving treatment adherence, they primarily focus on tracking medication intake rather than fostering interactive communication. MEMS and VOT, for example, provide adherence monitoring but may offer limited opportunities for patient-provider engagement. Two-way SMS messaging allows for basic interactions but often lacks flexibility for in-depth discussion or real-time problem-solving. In contrast, SM enables continuous, bidirectional communication, allowing patients to

seek clarifications, report challenges, and receive personalized support, which may be particularly beneficial for individuals facing adherence barriers beyond medication-taking behaviors [1,8,9].

Despite the growing adoptions of mobile health (mHealth) applications and DATs, there remains a significant gap in understanding how patients, particularly those from vulnerable populations, engage with these tools [10]. Individuals undergoing TB treatment face considerable challenges, including a demanding six-month treatment regimen that requires sustained adherence. While systematic reviews of DATs for TB care have shown promising yet mixed results, there is considerable variability in both the direction and magnitude of their impact on outcomes [7,11–16]. Most studies confirm the feasibility and acceptability of DATs, but there is a growing call to better understanding of patient engagement and its influence on treatment outcomes [7].

SM offers several advantages over traditional DATs by enhancing communication efficiency, improving patient-centered interactions, and fostering real-time problem solving [17,18]. SM enables dynamic secure information exchange which have been shown to improve adherence in other conditions, such as cancer and diabetes and has been reported to foster secure and timely responses, enhancing patient satisfaction and continuity of care [17–20]. However, evidence on its effectiveness in TB care and other chronic conditions remains limited, highlighting the need for further research.

This study aims to address this gap by analyzing SM interactions within a DAT intervention, the TB Treatment Support Tools (TB-TSTs) intervention, designed to support individuals with active TB during their treatment [21,22]. The TB-TST is a DAT that includes a mobile application with SM to connect patients with treatment supporters (TSs) and an objective adherence test. In a pragmatic randomized clinical trial conducted at four public reference hospitals in Argentina, the TB-TSTs intervention was found to improve treatment outcomes, including increased treatment success and reduce lost-to-follow up [23]. To better understand patient engagement, this study explores (a) how patients used SM to communicate with TB TSs, including the content and concerns raised, and (b) how TSs responded, such as issue resolution and proactive outreach. By examining these interactions, this research contributes to the digital health literature and provides insights for refining the TB-TSTs and other DATs to enhance their effectiveness in supporting medication adherence.

## Methods

### Study design

We conducted a retrospective descriptive qualitative study using thematic analysis to assess transmitted SMs in the Companion application as part of a larger pragmatic clinical trial assessing the effectiveness of the TB-TST intervention [23]. By better understanding patients' use of the intervention, we aimed to identify what worked and what to improve to meet users' needs and plan for scalability and sustainability. We coded all SMs between patients and TSs (255 communication channels). This manuscript has been structured following the guidelines set by the Consolidated Criteria for Reporting Qualitative Research (COREQ) [24].

### Study setting and participants

This study was conducted at four large public hospitals in Argentina, between November 2020 and May 2023, a period that included the COVID-19 pandemic and a 150-day national lockdown. These hospitals were selected due to their high TB patient volume and role as national reference centers for TB care. Participants were individuals randomized to the intervention group. Eligibility criteria included: 16 years of age or older, newly diagnosed with drug-susceptible TB, having regular access to a smartphone, and able to operate the phone or have someone available to assist [23]. Exclusion criteria were having severe illness requiring hospitalization, residence in the same household as another participant, or having confirmed drug-resistant TB. The study included six TSs - three pulmonary specialized physicians and three social workers from the recruitment sites. TSs received training on the application functionalities, treatment supporter dashboard, research objectives, and study protocols. For training and to identify any workflow issues, TSs

participated in field testing where over 10 days they alternated between patient and treatment supporter roles. This facilitated their ability to provide training and guidance to participants during study onboarding. Their primary responsibilities were to monitor daily reports, address participant inquiries, and facilitate communication to manage unforeseen events. By providing trusted guidance and real-time support, TSs aimed to enhance participant engagement and adherence to treatment.

### Intervention

The TB-TST intervention includes a mobile application, the Companion application, that connects users to TSs and a direct urine drug metabolite test to objectively confirm treatment adherence. The application, version 2.0+, is a progressive web application to report daily medication self-administration and side effects, learn about TB and its treatment, message TSs, view treatment progress, and interact with others in treatment through an anonymous discussion forum. The application's treatment supporter dashboard allows treatment supporters to monitor daily reports, assist patients with questions and challenges, and address issues. The Companion application has offline capabilities but requires Wi-Fi or mobile data to sync reports or send messages. TSs were responsible for training participants. This training took place at enrollment when the application was introduced, downloaded at the hospital, and included hands-on practice with its features. Supplemental material provided illustrated the application interface and features.

### Data collection

We collected all SMs sent from patients and TSs during the trial. The SM dataset contained participant study ID, the date and time the message was sent, the message text, and the TS study ID. The messages were dichotomized as patient-initiated or treatment supporter-initiated message. The TS response message was defined as a reply to a patient message. TS-initiated messages were messages to patients created *de novo* by the provider. The entire SM dataset contained patient-initiated messages, patient response messages, TS-initiated, and TS response messages.

### Data analysis

We used a multi-step iterative pragmatic approach, [25] combining both descriptive statistics and qualitative thematic analysis to explore the interactions between patients and TSs. Descriptive statistics were used to quantify communication patterns, including message frequency, sender type (patient or TS), and discussion topics such as symptoms or unrelated medical concerns. For the qualitative analysis, message transcripts were uploaded to Atlas.Ti (version 24.1.0.30612) for thematic coding in their original language, Spanish. Each message was categorized based on the sender type and discussion theme allowing for identification of key topics, message scenarios, and content trends throughout the treatment process.

To ensure rigor and reliability, we employed an iterative coding process and investigator triangulation. A subset of 15 message logs was independently coded by two trained researchers (JP and AV) and memos were kept documenting preliminary impressions and insights. The coding discrepancies were discussed in weekly meetings with the full research team to refine the coding categories. Once finalized, the coding framework was applied by six bilingual researchers (JP, EC, JR, AT, AV, and PC) trained in qualitative analysis. The diverse research team, with backgrounds in social and medical sciences, conducted repeated readings of the dataset to validate emerging themes, capturing both implicit and explicit ideas, while weekly discussions helped resolve discrepancies and refine thematic categories. Key illustrative quotations were translated into English, with bilingual researchers ensuring linguistic accuracy. From the analysis, we synthesized recommendations based on identified patterns and trends. Successful practices were derived from positive participant feedback, while areas for improvement were identified based on reported barriers and challenges. These recommendations were iteratively refined through team discussions to ensure they were grounded in the data and aligned with

the study's objectives. This combined analytical approach provided a comprehensive understanding of the interactions between patients and TSs, informing strategies to enhance TB treatment support.

## Ethical considerations

This study was approved by the Institutional Review Boards of the Ministry of Health of the province of Buenos Aires (ACTA-2019–15552860-GDEBA-CECMSALGP), the University of Washington institution review board Committee in Seattle, United States (STUDY00007533) and from the ethical committee of each participating hospital. All the participants provided written informed consent.

**Table 1. Patient baseline survey characteristics.**

| Characteristics | N (%) (277) |
|---|---|
| **Gender, n (%)** | |
| Female | 144 (52.0) |
| Male | 132 (47.7) |
| Non-binary | 1 (0.4) |
| **Age Group, years, n (%)** | |
| 16-24 | 84 (30.3) |
| 25-34 | 90 (32.5) |
| 35-44 | 48 (17.3) |
| 45-54 | 32 (11.6) |
| ≥55 | 23 (8.3) |
| **Level of Education, n (%)** | |
| Unable to Read or Write | 2 (0.7) |
| Primary Incomplete | 13 (4.7) |
| Primary Completed | 48 (17.3) |
| Secondary Incomplete | 80 (28.9) |
| Secondary Completed | 86 (31.0) |
| Post-Secondary Education | 47 (17.0) |
| No Answer Given | 1 (0.4) |
| **Steady Employment, n (%)** | |
| No | 134 (48.4) |
| Yes | 143 (51.6) |
| **Income Level, n (%)** | |
| Below Poverty | 43 (15.5) |
| Poverty | 127 (45.8) |
| Above Poverty | 82 (29.6) |
| No Answer | 25 (9.0) |
| **Relationship Status, n (%)** | |
| Single | 141 (50.9) |
| Married/ Stable Relationship | 119 (43.0) |
| Separated/ Divorced | 9 (3.2) |
| Widowed | 8 (2.9) |
| **Argentine National, n (%)** | |
| No | 64 (23.1) |
| Yes | 213 (76.9) |

## Results

A total of 6,805 messages were sent (2,926 by patients; 3,879 by TSs) that were associated with 277 unique patients. Table 1 provides the patient characteristics. Over half of the participants were female (52%), the majority were between the ages of 16 and 34 (62.8%), and nearly half had completed secondary education or higher (48%). Approximately half were steadily employed (51%), while most reported an income level at or below the poverty line (61.3%). Half of the participants were single (50.9%), and the majority were from Argentina (76.9%).

### Message content

Messages were categorized into five main broad content themes: living with TB (plus COVID-19 pandemic impact), medical and symptom guidance, healthcare coordination, treatment/intervention adherence, and technical issues, with eight subthemes for patients and 6 for treatment supporters (Table 2). The most common messages sent by patients were questions on managing symptoms and side effects (532, 38.0%) while the most frequent messages sent by TSs were reminders to patients to submit reports or check in (1404, 59.7%). A portion of unclassified SMS messages were confirmations of medication intake. Although these messages were important for supporting treatment adherence, they were repetitive and identical and did not form a distinct theme. The following describes the main themes and subthemes with exemplar quotes translated into English. The overlapping patient and TS themes are reported together.

### Living with TB (plus COVID-19 pandemic impact)

**Back to normal (Patient).** Patients often reached out to TSs with questions about how and when they could safely resume certain daily activities. Uncertainty about when it was safe for them or others to return to their regular routines

**Table 2. Thematic analysis table of patient-treatment supporter messages.**

| Theme | Subtheme | Description |
|---|---|---|
| Living with TB (plus COVID-19 pandemic impact) | Back to normal (Patient) | Returning to daily life after TB treatment; effects. |
| | Impact of COVID-19 (Both) | How the pandemic affected TB treatment, care access, and well-being. |
| | Answering TB questions (Treatment supporters) | Responding to patient concerns, providing general TB-related information. |
| Symptom Guidance | Support for symptom/side effects (Patient) | Seeking advice on symptoms and side effects of TB treatment. |
| | Symptom/side effect guidance (Treatment supporters) | Providing medical guidance on managing symptoms and side effects. |
| | Urgent request for assistance (Patient) | Reporting serious symptoms and emergencies, requesting immediate help. |
| | Seeking and providing emotional support (Both) | Exchanging messages offering or seeking emotional reassurance. |
| Healthcare Coordination | Requesting and providing healthcare coordination (Both) | Asking for help scheduling appointments and accessing medication. TSs facilitating coordination. |
| Treatment and Intervention Adherence | Reminding to submit adherence report (Treatment supporter) | Sending reminders for patients to submit treatment adherence reports. |
| | Barriers to submitting reports (Patient) | Mentioning difficulties in submitting adherence reports, such as technical and personal barriers. |
| | Strategies to support adherence (Treatment supporters) | Offering encouragement, strategies, and solutions to help patients adhere to treatment. |
| Technical Issues | Requesting technical support/ Resolving technical issues (Both) | Reporting or troubleshooting technical problems related to the platform. |

after a TB diagnosis and the start of treatment was a common concern. Many of their inquiries focused on integrating their treatment schedule into their daily lives. Additionally, some patients sought further clarification on the instructions provided by their physician, particularly regarding their treatment plan, dietary recommendations, and isolation measures. The following quotations illustrate these types of concerns:

> *Hello, good afternoon. I was recently diagnosed with tuberculosis and I wanted to ask you if I can eat pizza. Because the diet they gave me didn't specify it very well. Thank you very much. (18-years-old female).*

> *Good morning, I'm still testing negative. Could I skip a day of medication? Or if I drink alcohol, toast with a glass of cider and take the medication, will something happen to me? (44-year-old female).*

Patients also expressed significant concern about the wellbeing of their family and close contacts. They frequently reached out to the TSs for guidance on isolation measures, particularly regarding whether it was safe to engage in social gatherings, attend family gatherings, run errands, or engage in intimate relationships. Concerns about the potential transmission were common, especially among patients who lived with relatives. For example:

> *I'm doing better. My brother has a bad cough. We don't live together, but we share a bathroom. He's diabetic and medicated, but I'm worried about him. (51-year-old male)*

**Impact of COVID-19 (Patient).** Patients frequently sought advice on whether they could receive the COVID-19 vaccine while undergoing treatment for TB, expressing concerns about the potential interactions between the vaccine and TB medication and the safety and adverse effects of the vaccine during treatment. The continuity of TB treatment during the COVID-19 pandemic was another major area of concern, with patients discussing the difficulties they faced in adhering to their treatment due to pandemic-related disruptions, including hospitalizations. Additionally, patients inquired about the safety of engaging in daily activities with appropriate precautions, such as wearing a mask, particularly when they were not exhibiting COVID-19 symptoms.

> *P: I wanted to let you know that yesterday I did a covid test and I just got a positive result. I am having a little difficulty breathing. I get agitated very easily and the coughing is becoming very annoying. Is COVID going to affect me in my TB treatment? I wanted to know if it will complicate my treatment, or if I need to take any special care. (32-year-old female)*

> *TS: You have to continue your treatment! This is important! If you have respiratory difficulty, please consult your doctor in the emergency room. (TS #2)*

**Answering TB questions (TS).** TSs frequently addressed patients' concerns regarding the disease, its treatment, and the impact of COVID-19 on TB treatment. Common topics included medication dosage, frequency, missed doses, uncertainty about what to do after taking the wrong dose, and potential drug interactions. Inquiries also explored the possible effects of having COVID-19 or being vaccinated against it on TB treatment. These questions reflected a desire to avoid compromising their TB treatment or experiencing unwanted side effects. Inquiries outside of the TS's expertise were escalated to a specialist, and the TS followed up on their recommendations. TSs also used these communications to educate patients about how TB can affect the body, the consequences of non-adherence to isolation guidelines, specific isolation measures related to COVID-19, and the interpretation of test results.

> *Good morning! I accidentally took two Isoniazid 300mg. Is it dangerous? I weigh 40 kg. (33-year-old female)*

> *P: Doctor, I have another question. Can I have energy drinks during treatment? (18-year-old female)*

*TS: Avoid alcohol and high-calorie drinks at the start of treatment if you have digestive symptoms like reflux or pain. You can drink energy drinks if you need to but don't drink too much and keep an eye on your stomach. (TS #4)*

## Symptom guidance

The most common reason patients contacted TSs was to seek guidance on managing symptoms and side effects, with additional requests for urgent assistance or emotional support.

**Support for symptom/side effects (Patient).** Inquiries generally fell into symptom management and concerns about whether specific symptoms were normal. Patients reported a wide range of symptoms, the most common being gastrointestinal discomfort, pain, respiratory problems, and skin problems (Fig 1). Gastrointestinal issues were the most frequently reported, including nausea, vomiting, stomach pain, diarrhea, constipation, and changes in stool color. Other symptoms included night sweats, metallic taste, and general cold-like symptoms (unspecified). They primarily sought recommendations on how to manage these symptoms or determine if urgent care was necessary. In many cases, patients wanted to confirm whether their symptoms were expected side effects of the treatment, seeking reassurance to stay calm or avoid unnecessary hospital visits. Additionally, patients frequently provided updates to TSs on whether their symptoms had resolved or persisted.

*[In response to patient's message about feeling sick] Don't take the medication again today. Have a bland diet and take it again tomorrow. Take half the dose before breakfast, then have breakfast and wait two hours before taking the rest. This should help your stomach tolerate the medication better. (TS #1)*

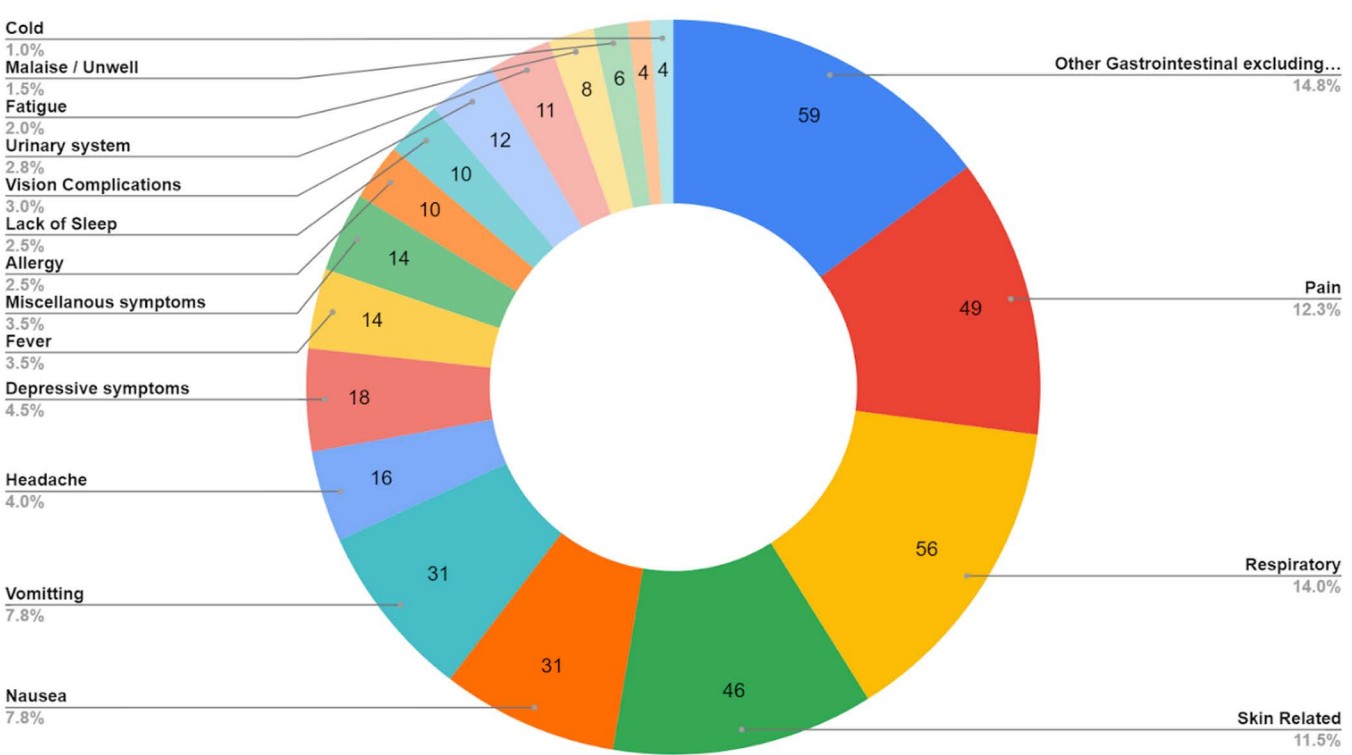

**Fig 1. Symptoms reported and frequency.**

*Good morning!! I wanted to ask a question: the muscle pain has not gone away. And now I have a black tongue. I feel a strong metallic taste when I eat everything and nausea. Should I continue with the medication? Should I go to the hospital? (43-year-old female)*

*Hi Doc! How's it going? I'm doing okay. The first day was a bit tough, with some discomfort, stomach pain, and nausea. But thankfully, I've been feeling much better since the second day. (30-year-old female)*

**Symptom/side effect guidance (TS).** TSs provided guidance on managing a wide range of symptoms, including respiratory issues, pain, skin conditions, mental health concerns and more. Their approach typically involved normalizing common symptoms associated with TB or its treatment, offering reassurance and helping patients avoid unnecessary anxiety or hospital visits. For example:

*The change in urine color is normal. The medication is orange, which can cause mucous membranes and secretions to change color during treatment. Don't be alarmed; it's just the medication. (TS#3)*

When patients presented with atypical or worsening symptoms, TSs conducted brief assessments of their condition. Based on these evaluations, TSs offered symptom management recommendations and scheduled follow-ups or referred patients to their treatment center or hospital for further care. In some cases, particularly involving skin conditions, TSs requested patients to send photographs to assist their decision-making process. The following exchange illustrates this type of interaction:

*P: Good morning, today in the report, I explained that I have hives. I have small welts on my legs that are red and very itchy. (26-year-old female)*

*TS: Good morning! Is it just one area or both entire legs? Can you send a photo?*

*P: Both legs, but especially from the knees up. (TS #1)*

*TS: Okay, show me a photo. I'll wait for it.*

*P: [sends Picture] It may not look very good, but they are small welts, like a rash.*

*TS: If it's only in the area around your knees, and you don't see it spreading to other body parts, it seems more like localized atopy. Pay attention to it so it doesn't affect different parts of your body and let me know!*

**Urgent request for assistance (Patient).** Patients frequently used the SM to report urgent health issues affecting themselves or their relatives. In these urgent situations, patients reported severe symptoms like respiratory distress, severe pain, persistent vomiting, and high fever.

*I'm getting worse and bleeding. I'm still taking my medication. I'm upset that they couldn't do an ultrasound. Please talk to the doctor; I need an urgent response. (22-year-old female)*

TSs promptly advised them to seek immediate medical attention at the nearest emergency room. Beyond providing immediate care advice, TSs played a role in coordinating care and follow-up care. This included providing instructions on managing medical consultations, emphasizing the importance of regular follow-ups with treating physicians and ensuring proper coordination for ongoing care. Sometimes, patients needed help navigating the healthcare system to arrange further medical attention, and the TSs were viewed as valuable advocates. Their support helped patients feel assured that they were receiving appropriate care and that all necessary steps were taken to address their urgent needs. The following quote exemplifies a patient experience:

**Seeking and providing emotional support (TS and Patient).** Emotional support emerged as a significant component of patient interactions with TSs. Many patients reported feelings of despair, anxiety, and uncertainty about their health situation, and for some, texting with TS served as a crucial emotional outlet.

*Hi, I'm having a panic attack, and I don't know what to do. I can't sleep. I don't know if that's what's giving me so much nausea. I'm only drinking liquids. I can't sleep at all, and I'm panicking. I feel like I'm going to die. (33-year-old female)*

This support became crucial for individuals who felt isolated during the prolonged COVID-19 lockdown—which overlapped with the study period—or who experienced a sense of being misunderstood during their primary care follow-ups. In response, TSs offered referrals to psychologists and psychiatrists and assisted with scheduling appointments, but they also closely monitored their emotional well-being. Beyond referrals, TSs provided practical strategies to manage negative emotions, such as recommending activities to keep the mind engaged and provide reassurance about their recovery process.

*P: I get very nervous.*

*TS: Try to calm down! Give it time! It's a lot of new drugs... the body has to assimilate them (TS #5).*

*That's great (name)! Yes, try to relax. While you can't go back to work look for some activity. Keeping your mind busy also helps a lot! (TS #4)*

Physical side effects of treatment, such as nausea, vomiting, dizziness, fatigue, and lack of appetite often heightened patients' emotional distress. Additionally, the stigma associated with TB amplified feelings of social isolation and fear, further exacerbating anxiety and emotional strain. Family dynamics also contributed to patients' emotional challenges. Many expressed concerns for their loved ones, such as one patient who was deeply worried about her husband's growing anxiety related to her illness and treatment. This combination of physical symptoms, social stigma, and family concerns created a complex emotional landscape for patients, where TSs played a vital role in offering support and assurance.

### Healthcare coordination

**Requesting and providing healthcare coordination (Patient and TS).** Patients frequently engaged with TSs to assist in coordinating hospital visits, rescheduling medical appointments, and navigating healthcare services. Appointment coordination was a common need, as patients often required assistance rescheduling or confirming appointments due to conflicts with their personal or work schedules, missed appointments, and other unforeseen circumstances. TSs provided essential administrative support, helping patients manage information related to their medical appointments, such as ensuring that contact numbers were readily available, documentation and records were in order, and that necessary steps were taken to avoid delays or disruptions in their care. Many patients expressed gratitude for the care provided at the hospital, largely facilitated by TSs. Patients often apologized for misunderstandings or communication difficulties, highlighting the value they placed on TSs and respect for their assistance.

*Hey there! I just wanted to let you know that I got the day off from work, so I'll be able to make it to the Dr. [name] consultation on Thursday the 15th. Sorry for the hassle, thanks a lot! (22-year-old female)*

### Treatment and intervention adherence

**Reminding to submit adherence report (TS).** Adherence to treatment and timely reporting of daily progress were critical components monitored by TSs throughout patients' TB treatment. To ensure consistent reporting and assess that the patients were on track, TSs frequently reached out to patients when their daily reports were missing. TSs varied in

their communication styles, which appeared to play a significant role in supporting intervention adherence. Some TSs used a personalized approach, tailoring their message to include the patient's name, expressing concern for their well-being, and showing empathy. These TSs often made individualized inquiries about the patient's condition which appeared to foster a stronger relationship, lead to greater patient engagement with the application, and prompt appreciative responses. For instance:

*Good afternoon [name]! I hope you are doing well and that the abdominal pain symptoms have subsided. I just wanted to remind you that the reports are daily. Please do not hesitate to contact me if I can help you with anything. Have a nice day! (TS #3)*

**One patient responded.** *Thank you very much for everything! Excellent service! I am forever grateful to all the staff of the hospital. (50-year-old female)*

Additionally, some TSs sent a welcome message that included information about the intervention, their availability, and application-related features. Over time, this introductory communication was sometimes updated to include additional instructions, such as those related to claiming phone credit. This initial contact set the tone for the ongoing interaction and ensured patients were well-informed about using the application effectively. Additionally, some TSs sent messages recognizing significant treatment milestones. Some used standardized messages to congratulate patients on reaching these milestones, such as completing a month of treatment or nearing the end of the treatment course. These messages often included next steps, like scheduling a final follow-up or interview. Others sent more personalized messages, acknowledging the patient's individual journey and expressing genuine pride in their progress. Such communication likely reinforced adherence by making patients feel more connected and supported throughout their treatment:

*Good morning! It's a pleasure to be your TS! I will accompany you during the treatment, answer questions, and help you in any way I can! I will respond to messages Monday to Friday from 8 am to 2 pm. After that, I will reply the next day! If you receive the message to activate the credit, we send you a monthly message to use the application and recharge your credit through SMS. Every month, we load credit so you can use the application! (TS #1)*

In contrast, some TSs relied on standardized, generic messages sent to all patients in the same way, regardless of the patient's responsiveness. These automated messages often failed to address the patient's unique circumstances or health concerns. In cases where patients did not respond, these TSs continued to send repetitive reminders without changing their communication strategy. This lack of personalization and empathy appeared to be associated with reduced patient interactions and, in some cases where the TSs response was delayed, led patients to stop using the application altogether, highlighting the importance of timely interactions, as explained by a patient:

*P: My treatment is going really well. I'm getting better every day! Regarding the application, I contacted you [TS] because I'd run out of medication on 16 Dec, and you replied on 28 Dec. I'd already gone to the hospital, and they'd given me the medication by then. So, I figured if there's no follow-up, the application isn't going to be helpful to me. That's why I stopped using it. (39 Years Old, Male)*

*TS: we're sorry; maybe there was a mistake with the application. (TS#1)*

**Barriers to submitting reports (Patient).** When patients were reminded by TSs to upload their daily reports, they often explained the delays. The most common reason cited was forgetfulness. Patients also provided various personal reasons and family situations for not submitting their reports, including being overwhelmed with work or daily tasks, not taking the medication because they were feeling unwell, or not using their mobile phones. Many patients mentioned

that technical issues posed significant challenges when submitting their reports. The most frequently reported problems included not receiving notifications or encountering application glitches. Some patients also expressed difficulties navigating the application or using the test strips, which hindered their ability to complete their reports.

*Good morning. I had forgotten because I had a very hard day and was going to upload it later, but I forgot. I will be more attentive. Thank you very much. (23-year-old male)*

Good morning. My session was closed, and I couldn't open it again, not even with the correct password. (22-year-old male)

Contextual factors were also identified as barriers to accessing the application's features and completing reports. The most common issues were related to internet access, such as weak signal strength, running out of data credits, and being unable to afford an additional data plan. Problems with mobile devices or power outages further interfered with uploading reports. A small group of patients cited a lack of treatment or test strips as a reason for not completing reports. These patients often requested refills during their interactions with TSs.

*Hello, good afternoon. I didn't have electricity yesterday, and my phone was dead. But I'll be more attentive, thanks. (23-year-old male)*

*Yes, I'm fine. It's just that sometimes I'm at work and I don't have a signal, and then I forget to log into the application because I'm not a big fan of using my phone. (23-year-old male)*

**Technical issues**

***Requesting technical support/Resolving technical issues (Patient and TS).*** Patients frequently exchanged messages with TSs to resolve the technical problems related to the application's functions and features and address challenges they faced during daily use. Common issues included difficulties with loading reports, encountering error messages when sending information, and problems with logging in or keeping the application active. Some users also struggled with finding the link to download the application, setting it up on a new device, and navigating the user interface. One of the recurring technical challenges involved device cameras, preventing them from taking and uploading photos of test strips required by the application.

*Hey, I'm having trouble uploading photos. I can't get the application to access my camera. It says I need to give it permission, but I'm not sure how. (16-year-old female)*

Additionally, some patients reported that their report data was deleted or not saved correctly, negatively affecting their monitoring and treatment adherence record accuracy. To address these problems, TSs provided detailed instructions on activating camera permissions, completing missing reports and uploading reports correctly. They also offered guidance on how to resolve login issues and how to navigate the application more effectively.

*P: I want to change my phone. How do I do it with the application? I looked for the application but couldn't find it. (33-Year-old female)*

*TS: Hi! Here's the link. You put it into Google: asistente.cirg.washington.edu. The blue start screen will appear as usual, and you will use your old phone number as your username and the same password.*

*P: Thank you!*

*TS: You're welcome! Let me know if you need anything else and I'll help you!*

Moreover, intervention-related communication often focused on managing the logistics of the monthly phone credit provided to patients to ensure they could access the application. This communication included addressing issues that arose when patients changed their phone or phone number, dealing with situations where the phone carrier did not recognize the patient's registered phone number, and responding to inquiries about when the credit would be granted.

Most interactions involved TSs informing patients that their phone credit had been provided on the previous day or the day of the message. In some cases, TSs provided a one-time redeem code to ensure the patient had sufficient credit. There were instances where patients needed to change their phone or phone number due to issues like poor battery life, which sometimes led to complications in receiving the credit. Additionally, when the phone carrier did not recognize the registered phone number, TSs needed to clarify the issue and gather further information from the patient to resolve it.

*Hello [patient's name], just insert the link [link] into your phone. There's the same login screen as your previous phone had, you will put your same username and add that this is your phone number even though using another one, also use the same password. (TS #2)*

**Key findings and recommendations to improve TB treatment support and DAT interventions**

Our analysis identified key insights into the effectiveness of treatment support and DAT interventions. By examining communication patterns and participant experiences, we synthesized recommendations to enhance the application for TB care. Successful practices were derived from positive participant feedback, highlighting aspects of the intervention that facilitated adherence and engagement. Conversely, areas for improvement emerged from reported barriers and challenges and identification of gaps, offering opportunities for refinement. These participant-driven recommendations are designed to inform the development of DAT tools and optimize support systems, ensuring that the intervention is patient-centered, responsive to needs, and effective in improving treatment adherence. By leveraging these findings, we can strengthen digital health strategies to enhance treatment experiences, address critical gaps, and health outcomes for individuals undergoing TB care.

> ## Key strengths to maintain and build upon
>
> - Personalize Communication: Tailor messages to individual patient needs, addressing specific health concerns, and incorporating motivational messages
>
> - Trust Building: Foster strong, supportive relationships between treatment supporters and patients
>
> - User-Friendly Adherence Reporting: Ensure the application remains intuitive and easy to use for tracking treatment progress
>
> - Personalized Reminders: Continue providing personalized reminders, which were well-received and appreciated by patients
>
> - Timely Information Sharing: Maintain a platform that allows patients to report and receive guidance on a wide range of symptoms
>
> - Expanded Mental Health Support: Strengthen mental health referrals and emotional well-being resources within the intervention
>
> - Streamlined Healthcare Coordination: Reduce administrative and logistical challenges in navigating healthcare services

- Technical Support: Continue offering problem-solving assistance for application use and intervention engagement
- Reassurance on Medication Side Effects: Enhance guidance on managing common TB treatment side effects, including emergency-use scenarios

## Areas for improvement and future enhancements

- Empathetic Communication Training: Strengthen TS training to enhance personalized, supportive, and empathetic patient engagement
- Clear Onboarding and Guidance: Improve welcome messages and guidance from all treatment supporters to ensure clarity and consistency
- Emergency protocols: Clearly define and communicate emergency response procedures
- Technical Enhancements: Address identified application glitches, simplify the interface further, and improve camera functionality
- Flexible Reporting Options: Offer multiple ways for patients to report adherence and concerns to accommodate different preferences and circumstances
- Enhance Connectivity Solutions: Explore alternative strategies to improve digital access with patients with limited connectivity
- Integrated Mental Health Services: Strengthen connection to mental health services and establish peer support groups
- Education Resources on TB: Expand educational content to address stigma and misinformation surrounding TB
- Systematic Issue Tracking: Improve intervention monitoring to ensure unresolved issues are escalated and resolved efficiently
- AI-Enhanced Treatment Support: To address patient request for 24/7 support, explore innovative strategies such as artificial intelligence (AI)-powered virtual TS to offer rapid tailored responses based on patient queries and prior interactions.
- Cost Evaluation: Assess intervention costs to inform sustainability and scalability
- Patient Success Stories: Incorporate testimonials from individuals who successfully completed TB treatment to inspire and encourage adherence

## Discussion

This study aimed to explore how a mobile application was utilized by individuals with TB and the role of TSs in addressing the challenges faced by patients. Patients frequently sought advice on integrating TB treatment into their daily routines, managing symptoms and side effects, medication interactions, and coping with the isolation associated with the disease. A prominent concern for many patients was the safety of daily activities, especially in the context of the COVID-19 pandemic. Patients frequently inquired about TB transmission, vaccine safety, and the pandemic's potential impact on their treatment continuity. These inquiries underscore the critical need for accurate medical advice and emotional support to promote sustained treatment adherence. Additionally, many patients required assistance in coordinating healthcare appointments, addressing technical issues, and navigating personal circumstances, emphasizing the multifaceted role of TSs in supporting patients beyond simply reminding them to submit daily reports.

The primary goal of the application and SM feature, consistent with existing evidence, was to improve treatment adherence [13,26–29]. Mobile applications have been shown to be effective in facilitating early screening, diagnosis, and treatment, thereby preventing health complications and advancing preventive medicine [29]. In our study, TSs frequently used the application's SM feature to prompt patients to report their medication intake and address related challenges. However, patients often leveraged the messaging feature for additional purposes beyond simply reporting. Consistent with findings from other mHealth interventions [29–31], patients sought guidance on symptom management, treatment side effects, and potential medication interactions. This highlights the value of real-time communication in delivering personalized, timely information and the importance of flexibility in mHealth tools to meet patients´ diverse needs.

Another important finding was the variation in communication styles among TSs. Some TSs used personalized, empathetic approaches, while others relied on standardized, generic messages. The more personalized communication strategies often led to better patient engagement and satisfaction, while impersonal, repetitive reminders or delayed responses sometimes resulted in reduced interaction and even disengagement. Previous studies have highlighted the importance of warm, respectful communication in clinical interactions, noting that these qualities contribute to better health outcomes [32]. In our study, most TSs maintained a professional yet approachable tone in their messages, using clear, colloquial language to engage patients. This type of interaction fosters person-centered care, which is essential in complex care situations [31,32].

Patients used the SM not only to seek emotional support but also to request assistance in scheduling appointments with mental healthcare providers. While it was expected that TB-related stigma and guilt often reported in the literature, would lead some patients to seek emotional reassurance, the extent of the need for mental health support was somewhat surprising. Many patients expressed significant emotional distress related to their TB diagnosis, including feelings of isolation, stigma, and uncertainty underscoring the unmet demand for integrated mental health care. The SM feature provided a safe and non-threatening platform for patients to share their experiences and seek emotional support from TSs. These findings align with previous research demonstrating that mHealth tools can facilitate patient-provider connections and help patients manage mental health challenges, particularly during stressful events like the COVID-19 pandemic [4,33]. However, the high volume of mental health-related concerns highlights an urgent need to enhance TB care with dedicated psychological support and referral pathways [34–36].

Technical issues with the application, including difficulties with report submissions, login or application glitches as well as intermittent access to Wi-Fi or data frequently hindered patients' ability to engage with the intervention. These technical challenges reflect broader issues seen in other mHealth interventions in Latin America, where poor infrastructure, unreliable internet connectivity, and limited access to electricity are common [37]. Moreover, low technological literacy among some patients likely exacerbated these challenges, further hindering their engagement with the interventions [29].

Our findings have implications for the practical application and development of mHealth interventions. The application's ability to motivate patients, facilitate digital reporting, and provide real-time support highlights the importance of incorporating personalized and supportive elements to improve intervention and treatment adherence. Improving the responsiveness and personalization by TSs could significantly improve patient engagement and satisfaction. Given the frequent requests for advice on symptom management, treatment side effects, drug interactions and emotional support, TSs need to be trained to provide accurate advice and emotional support. Additionally, addressing technical bugs, ensuring application stability, and maintaining connectivity are fundamental to providing a reliable user experience. Finally, the application's role in coordinating healthcare appointments highlights the need for efficient and responsive communication channels within mHealth tools. Improving these aspects could maximize the effectiveness of mHealth interventions for TB and other chronic diseases, ensuring patients receive the comprehensive support they need to adhere to their treatment regimens.

## Limitations and strengths

The study has several limitations that should be acknowledged. First, it focuses on individuals with drug-susceptible TB who had access to smartphones, which limits the generalizability of the findings to other groups, such those with

drug-resistant TB or without access to smartphones. Additionally, as the study was conducted in Argentina, its findings may not be fully translated to other socioeconomic or geographic contexts, particularly in countries with different health-care systems, digital infrastructure, or patient demographics. Future research should explore testing in diverse settings to assess its broader applicability. Another limitation is that the study did not directly analyze variations in TS response times, which may have influenced patient satisfaction and perceptions of support effectiveness. Furthermore, differences in patients' digital literacy likely impacted their ability to use the application effectively, potentially skewing engagement patterns toward those more comfortable with technology. Future research should consider strategies to support patients with lower digital literacy to enhance equitable access. Additionally, translating participant messages from Spanish to English presents a risk of interpretation bias. Although steps were taken to mitigate this risk, including having bilingual researchers analyze the data in its original language and cross-checking translations to ensure accuracy, some nuances may have been lost in translation.

Despite these limitations, the study has several notable strengths. A key strength is the use of complete chat logs, providing a rich dataset for in-depth thematic analysis of patient-TS interactions. This comprehensive data set allowed for a nuanced understanding of how patients engaged with the application and how TSs responded to their needs. This study was also conducted in a real-world, pragmatic setting, reflecting actual usage patterns and enhancing the relevance of the findings. Moreover, the analysis was conducted in the original language by a bilingual team using established qualitative methods, ensuring both linguistic accuracy and a culturally sensitive interpretation of the data [38,39].

## Conclusion

This study provides insights into the use of a mobile application designed to support individuals during their TB treatment, highlighting the potential of mHealth interventions to improve patient adherence, provide timely medical advice, and offer emotional support. The application successfully facilitated communication between patients and TSs, allowing patients to seek guidance with treatment management, side effect mitigation, emotional distress, and challenges brought on by the COVID-19 pandemic. The variations in communication styles among TSs highlighted the importance of personalized and empathetic interactions in maintaining patient engagement and adherence. Future mHealth interventions should prioritize technological reliability, the human aspects of TB treatment and personalized care. Interventions should be adaptable, user-friendly, and capable of addressing the diverse needs of patients, including those with limited digital literacy. By improving these areas, mHealth interventions can support treatment adherence and contribute to improved health outcomes for TB and other chronic diseases.

## Supporting information

**S1 File. Inclusivity-in-global-research-questionnaire.**
(DOCX)

## Author contributions

**Conceptualization:** Fernando Rubinstein, Sarah Iribarren.

**Data curation:** Priscilla Carmiol-Rodriguez, Omar Alfonso Aguilar-Vidrio, Agnes A. Suyanto, Enrique Chan-Liang, Sarah Iribarren.

**Formal analysis:** Javier Roberti, Priscilla Carmiol-Rodriguez, Omar Alfonso Aguilar-Vidrio, Enrique Chan-Liang, Jennifer Sprecher.

**Funding acquisition:** Fernando Rubinstein, Sarah Iribarren.

**Investigation:** Javier Roberti, Priscilla Carmiol-Rodriguez, Omar Alfonso Aguilar-Vidrio, Agnes A. Suyanto, Enrique Chan-Liang, Jennifer Sprecher, Fernando Rubinstein, Sarah Iribarren.

**Methodology:** Omar Alfonso Aguilar-Vidrio, Jennifer Sprecher, Fernando Rubinstein, Sarah Iribarren.

**Project administration:** Sarah Iribarren.

**Resources:** Agnes A. Suyanto, Jennifer Sprecher.

**Software:** Omar Alfonso Aguilar-Vidrio, Agnes A. Suyanto, Fernando Rubinstein.

**Validation:** Agnes A. Suyanto, Jennifer Sprecher, Sarah Iribarren.

**Visualization:** Javier Roberti, Priscilla Carmiol-Rodriguez, Omar Alfonso Aguilar-Vidrio, Enrique Chan-Liang.

**Writing – original draft:** Javier Roberti, Priscilla Carmiol-Rodriguez, Omar Alfonso Aguilar-Vidrio, Enrique Chan-Liang, Sarah Iribarren.

**Writing – review & editing:** Javier Roberti, Priscilla Carmiol-Rodriguez, Omar Alfonso Aguilar-Vidrio, Agnes A. Suyanto, Enrique Chan-Liang, Jennifer Sprecher, Fernando Rubinstein, Sarah Iribarren.

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
