## [Decision Letter · Decision Letter 0]

Dear Dr. Iribarren,

We look forward to receiving your revised manuscript.

Kind regards,

Rogie Royce Carandang, RPh, MPH, MSc, PhD

Academic Editor

PLOS ONE

Journal Requirements:

“This study was funded by the United States National Institute of Health, National Institute of Allergy and Infectious Diseases (R01AI147129: Iribarren, Rubinstein), and is building on the development research funded by the National Institute of Nursing Research (K23NR017210: Iribarren).”

5. We note that you have indicated that there are restrictions to data sharing for this study. PLOS only allows data to be available upon request if there are legal or ethical restrictions on sharing data publicly. For more information on unacceptable data access restrictions, please see http://journals.plos.org/plosone/s/data-availability#loc-unacceptable-data-access-restrictions.

Additional Editor Comments:

**P6 L98-106:**  Do you have any figures or previously published papers that show or describe the app in more detail? Even though this is not about app development, you can add supplementary files that include screenshots of the app and its features.

**P7 L100-113:**  Please provide justification for the selection of the facilities. Why were these four hospitals chosen out of all the hospitals in Argentina?

**P8 L135-137:**  Please describe in more detail how rigor is ensured in the study. What specific strategies are employed by the team?

**Table 1:**  You mentioned that there are 255 unique patients, but Table 1 shows 277. Could you please clarify this discrepancy? Additionally, there is a superscript next to the number 55, but I could not find the corresponding footnote.

**Table 2:**  Please reorganize this table. It currently lacks essential details, such as the sub-themes. For example, what specific barriers did patients report under “Reporting barriers to submit reports”? As it stands, the table does not clearly present a thematic analysis.

**Quotes:**  I suggest indicating a description of who the treatment supporters are, similar to what you did with the patients. This way, readers will know from whom the advice is typically coming.

**P24 L439-445:**  Please indicate whether the recommendations were participant-driven, researcher-driven, or both. The process for arriving at these recommendations is not clear.

**L27 P469:**  You mentioned that “A somewhat unexpected finding was patients using SM to seek emotional support.” Please specify if this study was conducted during the middle of the COVID-19 pandemic, as this may help explain the results.

**P29 L512-514:**  Was there any prior training conducted for both patients and treatment supporters regarding the app? This detail was not mentioned in the methods section. I believe it is crucial to provide prior training on the app’s use to ensure better engagement.

**References:**  Please ensure that the revised manuscript follows the correct citation format.

Reviewers' comments:

Reviewer's Responses to Questions

**Comments to the Author**

1. Is the manuscript technically sound, and do the data support the conclusions?

Reviewer #1: Yes

Reviewer #2: Partly

Reviewer #3: Yes

2. Has the statistical analysis been performed appropriately and rigorously?

Reviewer #1: No

Reviewer #2: Yes

Reviewer #3: N/A

3. Have the authors made all data underlying the findings in their manuscript fully available?

Reviewer #1: Yes

Reviewer #2: Yes

Reviewer #3: Yes

4. Is the manuscript presented in an intelligible fashion and written in standard English?

Reviewer #1: Yes

Reviewer #2: Yes

Reviewer #3: Yes

Reviewer #1: The manuscript, "Digital Technology to Support Tuberculosis Treatment: Thematic Analysis of Interactive Messages to Understand Needs, Challenges, and Strategies to Improve Support Tools," provides a timely and valuable exploration of how digital tools can enhance the management of tuberculosis treatment.

Tuberculosis(TB) is still the world's top infectious killer according to WHO. Utilizing technology to promote medical adherence to TB is quite new and needs comprehensive understanding how this tool could provide better outcomes and adherence. With the increasing use of mobile phones, Secure Messages is like hitting two birds in one stone. It shows as a practical medium that address patient's needs and improve treatment adherence.

Suggestions for improvement:

Study Title: The title is more suitable for the short one since the study focused on Secure message threads only. Since digital technology might also refers to video observed therapy and digital pill based strategies.

Abstract:

Introduction: Kindly elaborate the process and advantages of SM compared to other DATs. Provide studies where SM has been used and is successful.

Population: Expanding it's socioeconomic population (eg low-income countries to high-income countries)for generalized findings. Since the study is just conducted to Argentina.

Message Content: The messages were only categorized into five main content themes. But the messages sent by patients are 2926 in total. The top 5 only has 1,397messages which is merely half of what it sent. Look further into other content for the application improvement.

Include the TB treatment supporters (TS) insights to provide their views and refinement while using the SM threads.

Kindly provide results findings table for better visualization.

Limitations: kindly provide risk of biases like interpretation of results since it was being translated from Spanish to English.

Overall the manuscript explored how the app was utilized by patients with TB. The app achieved it's goal to motivate patients, provide support to improve their treatment adherence.

Reviewer #2: I am a Novice to research review but Thank you for the opportunity to review Dr. Sarah J. Iribarren paper entitled “Digital Technology to Support Tuberculosis Treatment: Thematic Analysis of Interactive Messages to Understand Needs, Challenges, and Strategies to Improve Support Tools”. I hope these comments of mine will perhaps help this research.

Reviewer #3: First of all, thank you for inviting me to review the manuscript. The authors have qualitatively explored the factors that impacts the effectiveness of digital technologies such as secure messaging in the management of Tuberculosis (TB). Their findings shows that the manner treatment supports respond to the patients' inquiry and concerns play a huge role on the effectiveness of such digital technologies on the monitoring and treatment of TB. The manuscript is well-written and organize. I just have a minor comment on the use the "app" abbreviation. If possible, maybe the authors can use the unabbreviated word instead. On the discussion part, I like how the authors highlighted how findings from other studies relates to theirs and what it implies, although they might consider making the first paragraph a summary of the study's major findings alone with no citations yet then discuss these findings on the succeeding paragraphs starting with the most important to the least important.

**Do you want your identity to be public for this peer review?** For information about this choice, including consent withdrawal, please see our Privacy Policy

Reviewer #1: **Yes: ** Geselle Ann Guerrero

Reviewer #2: No

Reviewer #3: No

---

## [Author Response · Author response to Decision Letter 1]

8 Apr 2025

PONE-D-24-51272

Digital Technology to Support Tuberculosis Treatment: Thematic Analysis of Interactive Messages to Understand Needs, Challenges, and Strategies to Improve Support Tools

Rogie Royce Carandang, RPh, MPH, MSc, PhD

Academic Editor

PLOS ONE

Dear Dr. Royce Carandang,

Thank you to you and the reviewers for taking the time to provide detailed feedback on our manuscript. Your comments have improved our work. Please find below our detailed point-by-point responses to all the reviewers’ comments, along with the corresponding revisions made to the manuscript.

Editors’ comments

Please include a complete copy of PLOS’ questionnaire on inclusivity in global research in your revised manuscript. https://journals.plos.org/plosone/s/best-practices-in-research-reporting.

Response: We thank the editor for this comment. We are including this questionnaire.

Note 1. financial disclosure: Please note that funding information should not appear in any section or other areas of your manuscript.

Thank you for stating the following financial disclosure: This study was funded by the United States National Institute of Health, National Institute of Allergy and Infectious Diseases (R01AI147129: Iribarren, Rubinstein), and is building on the development research funded by the National Institute of Nursing Research (K23NR017210: Iribarren). Please state what role the funders took in the study. If the funders had no role, please state: "The funders had no role in study design, data collection and analysis, decision to publish, or preparation of the manuscript." Please include this amended Role of Funder statement in your cover letter; we will change the online submission form on your behalf.

Response: We thank the editor for this comment. We have added this phrase.

Note 2. Data sharing: We note that you have indicated that there are restrictions to data sharing for this study. PLOS only allows data to be available upon request if there are legal or ethical restrictions on sharing data publicly. For more information on unacceptable data access restrictions, please see http://journals.plos.org/plosone/s/data-availability#loc-unacceptable-data-access-restrictions. Before we proceed with your manuscript, please address the following prompts: a) If there are ethical or legal restrictions on sharing a de-identified data set, please explain them in detail (e.g., data contain potentially identifying or sensitive patient information, data are owned by a third-party organization, etc.) and who has imposed them (e.g., a Research Ethics Committee or Institutional Review Board, etc.). Please also provide contact information for a data access committee, ethics committee, or other institutional body to which data requests may be sent. b) If there are no restrictions, please upload the minimal anonymized data set necessary to replicate your study findings to a stable, public repository and provide us with the relevant URLs, DOIs, or accession numbers. For a list of recommended repositories, please see https://journals.plos.org/plosone/s/recommended-repositories. You also have the option of uploading the data as Supporting Information files, but we would recommend depositing data directly to a data repository if possible. We will update your Data Availability statement on your behalf to reflect the information you provide.

Response: We have indicated that there are restrictions to data sharing for this study because data contain potentially identifying or sensitive patient information.

Note 3. Please include captions for your Supporting Information files at the end of your manuscript, and update any in-text citations to match accordingly. Please see our Supporting Information guidelines for more information: http://journals.plos.org/plosone/s/supporting-information.

Response: NA

Additional Editor Comments:

Comment 1. P6 L98-106: Do you have any figures or previously published papers that show or describe the app in more detail? Even though this is not about app development, you can add supplementary files that include screenshots of the app and its features.

Response: We thank the reviewer for this suggestion. We have added previous publications that describe the process of app development in more details. We also included supplemental figures showing interface of app [1, 2].

Comment 2: P7 L100-113: Please provide justification for the selection of the facilities. Why were these four hospitals chosen out of all the hospitals in Argentina?

Response: We thank the reviewer for this comment. These facilities were selected because they treat the highest number of TB patients in the country and serve as national reference centers for tuberculosis. We have incorporated this information into the methods section in Study setting and participants.

Comment 3: P8 L135-137: Please describe in more detail how rigor is ensured in the study. What specific strategies are employed by the team?

Response: Thank you for your comment. We have added information in the methods section describing our strategies. To ensure rigor, we employed investigator triangulation, systematic coding procedures, and ongoing team discussions. Initially, two researchers independently coded a subset of message logs, comparing results and refining the coding framework through discussions. Weekly team meetings facilitated the resolution of discrepancies, refinement of thematic categories, and documentation of emerging insights. Once finalized, the coding framework was systematically applied by six bilingual researchers, ensuring consistency and validity. Regular discussions and repeated readings of the dataset further validated thematic categories. Key illustrative quotations were translated into English, with bilingual researchers ensuring accuracy.

Comment 4: Table 1: You mentioned that there are 255 unique patients, but Table 1 shows 277. Could you please clarify this discrepancy? Additionally, there is a superscript next to the number 55, but I could not find the corresponding footnote.

Response: We thank the reviewer for pointing this out. This has been corrected. The superscript was a ‘greater than’ symbol which was wrongly changed.

Comment 5: Table 2: Please reorganize this table. It currently lacks essential details, such as the sub-themes. For example, what specific barriers did patients report under “Reporting barriers to submit reports”? As it stands, the table does not clearly present a thematic analysis.

Response: We thank the reviewer for this thoughtful comment. We have reworked Table 2 to include a short description of each subtheme, providing greater clarity and detail about the thematic analysis. However, to maintain the table’s readability and avoid redundancy, we have limited the level of detail included in the table itself. Additional information about specific barriers, such as those reported under “Reporting barriers to submit reports,” is thoroughly discussed in the main text. We believe the revised table now serves as a useful overview of the findings, presenting the themes, subthemes, and their content in a clear and concise manner.

Comment 6: Quotes: I suggest indicating a description of who the treatment supporters are, similar to what you did with the patients. This way, readers will know from whom the advice is typically coming.

Response: We thank the reviewer for this suggestion. We have added information in the main text. The six treatment supporters (TSs) consisted of three pulmonary specialized physicians and three social workers from the recruitment sites. We could not provide additional details on who said each quote in the paper because the small number of treatment supporters in our study makes it difficult to maintain confidentiality and anonymity. Including such information could inadvertently lead to identification, which would compromise the ethical principles of protecting participant privacy.

Comment 7: P24 L439-445: Please indicate whether the recommendations were participant-driven, researcher-driven, or both. The process for arriving at these recommendations is not clear.

Response: We appreciate the reviewer’s suggestion and have incorporated the requested information in the Methods section and the introductory paragraph before the recommendations. The recommendations, based on qualitative analysis of participant messages, are participant-driven, came from treatment supporters or patients reflecting strengths, gaps, and challenges from the trial. Where applicable, we specify whether insights were from treatment supporters or patient participants.

Comment 8: L27 P469: You mentioned that “A somewhat unexpected finding was patients using SM to seek emotional support.” Please specify if this study was conducted during the middle of the COVID-19 pandemic, as this may help explain the results.

Response: We thank the reviewer for this valuable comment. We clarified our finding that although seeking mental health/emotional support was expected, the extent of the need for mental health support from the treatment supporters was somewhat surprising. We incorporated the suggested information in both the Methods and Results sections, where this point is addressed. Throughout the manuscript, the impact of the COVID-19 pandemic is highlighted multiple times, as it significantly affected patients and healthcare services. This was particularly relevant in the context of Argentina, where the 150-day lockdown—the longest in the world—further exacerbated these challenges. We also added references to the literature highlighting the importance of addressing mental health within TB care.

Comment 9: P29 L512-514: Was there any prior training conducted for both patients and treatment supporters regarding the app? This detail was not mentioned in the methods section. I believe providing prior training on the app’s use is crucial to ensure better engagement.

Response: We thank the reviewer for this important observation. We have now included details about the training process in the methods section to provide clarity. Treatment supporters received comprehensive training on the app’s use, after which they were responsible for training patients. Treatment supporters went through field testing where they acted as patients and then as treatment supporters to gain an understanding of the app and how it would be used by the patients. We added this to the methods section. Then, treatment supporters helped patients download the app at the hospital and conducted a hands-on practice session in the office. We agree that prior training is crucial for ensuring better engagement, and we appreciate the opportunity to clarify this aspect of the study.

Comment 10: References: Please ensure that the revised manuscript follows the correct citation format.

Response: We thank the reviewer for this comment. We have revised all citations and corrected when necessary.

Reviewer #1:

Comment 11: Study Title: The title is more suitable for the short one since the study focused on Secure message threads only. Since digital technology might also refers to video observed therapy and digital pill based strategies.

Response: We thank the reviewer for this insightful comment. We agree that the shorter title, 'Analyzing Messages to Enhance TB Treatment Support,' more accurately reflects the study's focus on secure message threads. We have therefore revised the title accordingly to better align with the study's specific focus and findings.

Comment 12: Introduction: Kindly elaborate the process and advantages of SM compared to other DATs. Provide studies where SM has been used and is successful.

Response: Thank you for your comment. We have addressed this suggestion in the introduction. We added a paragraph comparing other DATs to SM.

Comment 13: Population: Expanding it's socioeconomic population (eg low-income countries to high-income countries) for generalized findings. Since the study is just conducted to Argentina.

Response: Thank you for your suggestion. We agree that expanding the study population to include a broader socioeconomic range would enhance the generalizability of the findings. We recognize this as a limitation and have included a discussion in the manuscript about how the findings may be context-specific and less generalizable to other socioeconomic settings.

Comment 14: Message Content: The messages were only categorized into five main content themes. But the messages sent by patients are 2926 in total. The top 5 only has 1,397messages which is merely half of what it sent. Look further into other content for the application improvement.

Response: Thank you for your comment. We appreciate your observation regarding the proportion represented by the top five content themes. To clarify, the discrepancy in the numbers arises from the way messages were counted and coded during analysis. While individual messages were counted separately (e.g., a treatment supporter sending "Please send report" multiple times in a row would count as three individual messages), during the coding process, such repetitive messages—when they appeared consecutively and conveyed the same intent—were grouped and coded as a single unit of analysis (e.g., one instance of "treatment supporter reminding the patient to report"). This approach was taken to avoid overcounting repetitive content and to ensure the coding reflected meaningful thematic patterns rather than sheer volume. We added this information in the text: A portion of unclassified SMS messages served as confirmations of medication intake. Although these messages were important for supporting treatment adherence, they were repetitive and identical and did not form a distinct theme.

Comment 15: Include the TB treatment supporters (TS) insights to provide their views and refinement while using the SM threads.

Response: Thank you for your insightful comment. This study includes themes from both patient and treatment supporter messages. Based on these findings recommendations for refinement were synthesized. As a separate study, reported elsewhere, we conducted interviews with patients and treatment supporters to explore perspectives and experiences.

Comment 16: Kindly provide results findings table for better visualization.

Response: We thank the reviewer for this suggestion. In response, we have included a detailed table summarizing the key themes and subthemes derived from the qualitative analysis, which provides a clear and structured visualization of the findings.

Comment 17: Limitations: kindly provide risk of biases like interpretation of results since it was being translated from Spanish to English.

Response: We thank the reviewer for raising this important point. We have now included a discussion of potential biases, particularly those related to the interpretation of results arising from the translation of data from Spanish to English. We highlight that the analysis was conducted in the original language, Spanish by bilingual research team members. Only the example quotes were translated. Nonetheless, risk for bias in interpretation of results has been added to the Limitations section, where we acknowledge the steps taken to mitigate these risks.

Reviewer #3:

Comment 18: The manuscript is well-written and organize. I just have a minor comment on the use the "app" abbreviation. If possible, maybe the authors can use the unabbreviated word instead.

Response: We thank the reviewer; we have changed app to application throughout the text.

Comment 19: On the discussion part, I like how the authors highlighted how findings from other studies relates to theirs and what it implies, although they might consider making the first paragraph a summary of the study's major findings alone with no citations yet then discuss these findings on the succeeding paragraphs starting with the most important to the least important.

Response: We have revised the Discussion section by reorganizing and relocating specific lines to improve the flow and clarity of the discussion and reflection on relevant existing literature.

---

## [Decision Letter · Decision Letter 1]

Enhancing Tuberculosis Treatment Support: A Thematic Analysis of Interactive Messages in a Digital Adherence Technology Trial to Identify Needs, Challenges, and Strategies for Improvement

PONE-D-24-51272R1

Dear Dr. Iribarren,

We’re pleased to inform you that your manuscript has been judged scientifically suitable for publication and will be formally accepted for publication once it meets all outstanding technical requirements.

Kind regards,

Rogie Royce Carandang, RPh, MPH, MSc, PhD

Academic Editor

PLOS ONE

Additional Editor Comments (optional):

Reviewers' comments:

Reviewer's Responses to Questions

**Comments to the Author**

Reviewer #1: All comments have been addressed

Reviewer #2: All comments have been addressed

Reviewer #3: All comments have been addressed

2. Is the manuscript technically sound, and do the data support the conclusions?

Reviewer #1: Yes

Reviewer #2: Yes

Reviewer #3: Yes

3. Has the statistical analysis been performed appropriately and rigorously?

Reviewer #1: Yes

Reviewer #2: Yes

Reviewer #3: Yes

4. Have the authors made all data underlying the findings in their manuscript fully available?

Reviewer #1: Yes

Reviewer #2: Yes

Reviewer #3: Yes

5. Is the manuscript presented in an intelligible fashion and written in standard English?

Reviewer #1: No

Reviewer #2: Yes

Reviewer #3: Yes

Reviewer #1: My initial comments have been addressed. I only have one comment on line 541: please insert a space in the word "itfocuses".

Reviewer #2: I appreciate the authors’ thorough revisions and responses to my comments. All my concerns have been adequately addressed, and the manuscript has been significantly improved. I have no further suggestions for changes.

Reviewer #3: The topic of the study is very interesting as Digital Adherence Technology is something that is new to me, and I think it is a great innovation to support treatment of Tuberculosis. The introduction clearly presents the background of the topic, the problem is well identified, and the objective is clear. The method used is explained and the conclusions support the results of the analysis.

**Do you want your identity to be public for this peer review?** For information about this choice, including consent withdrawal, please see our Privacy Policy

Reviewer #1: No

Reviewer #2: No

Reviewer #3: No

---

## [Editor Report · Acceptance letter]

PONE-D-24-51272R1

PLOS ONE

Dear Dr. Iribarren,

I'm pleased to inform you that your manuscript has been deemed suitable for publication in PLOS ONE. Congratulations! Your manuscript is now being handed over to our production team.

Kind regards,

on behalf of

Dr. Rogie Royce Carandang

Academic Editor

PLOS ONE